# The Analysis of Human Serum N-Glycosylation in Patients with Primary and Metastatic Brain Tumors

**DOI:** 10.3390/life11010029

**Published:** 2021-01-06

**Authors:** Csaba Váradi, Viktória Hajdu, Flóra Farkas, Ibolya Gilányi, Csaba Oláh, Béla Viskolcz

**Affiliations:** 1Institute of Chemistry, Faculty of Materials Science and Engineering, University of Miskolc, 3515 Miskolc, Hungary; kemviki@uni-miskolc.hu (V.H.); bela.viskolcz@uni-miskolc.hu (B.V.); 2Borsod-Abaúj-Zemplén County Center Hospital and University Teaching Hospital, 3526 Miskolc, Hungary; vfflora@bazmkorhaz.hu (F.F.); gilanyi.lab@bazmkorhaz.hu (I.G.); olah.idegseb@bazmkorhaz.hu (C.O.)

**Keywords:** serum glycosylation, brain tumor, liquid chromatography, mass spectrometry

## Abstract

The identification of patients with different brain tumors is solely built on imaging diagnostics, indicating the need for novel methods to facilitate disease recognition. Glycosylation is a chemical modification of proteins, reportedly altered in several inflammatory and malignant diseases, providing a potential alternative route for disease detection. In this paper, we report the quantitative analysis of serum N-glycosylation of patients diagnosed with primary and metastatic brain tumors. PNGase-F-digested and procainamide-labeled serum glycans were purified by magnetic nanoparticles, followed by quantitative liquid chromatographic analysis. The glycan structures were identified by the combination of single quad mass spectrometric detection and exoglycosidase digestions. Linear discriminant analysis provided a clear separation of different disease groups and healthy controls based on their N-glycome pattern. Altered distribution of biantennary neutral, sialylated but nonfucosylated, and sialylated–fucosylated structures were found to be the most significant changes. Our results demonstrate that serum glycosylation monitoring could improve the detection of malignancy.

## 1. Introduction

The incidence of brain and central nervous system (CNS) cancers has been more prevalent in recent years, causing significant morbidity and mortality worldwide [1]. In 2016, 330,000 brain and CNS cancer-related cases and 227,000 deaths were reported globally, with the highest numbers in China, the USA, and India [1]. Children and adults can both be affected by CNS cancers, starting with heterogeneous symptoms that depend on the location of the tumors [2]. First signs can include headaches, vision loss, speech disturbance, and, in more severe cases, seizures and paresis [3]. Histologically, brain tumors can originate from different cells of the CNS and also from other organs through metastasis [4]. The most common type of primary brain tumor is meningioma, which is a slow-growing tumor originating from the arachnoid cap cells, often causing no symptoms over years [5]. Meningiomas are more frequent in women, and ~90% of the cases are benign, although they still can cause significant problems for the patients [6]. The second most frequent primary brain tumors arise from glial cells, resulting in high-grade (glioblastomas) and low-grade gliomas (astrocytoma, oligodendroglioma) [7]. Glioblastomas are the most aggressive cancerous tumors, with a low survival rate, usually causing death within two years of diagnosis despite surgical and oncological treatments [8]. Brain tumors can also be metastatic due to the spreading of cancer cells from their original site. Any type of cancer can develop metastasis in the brain, although the most common ones are melanomas, lung, breast, renal, and colon cancers [9]. One of the main problems with brain tumors is the lack of body fluid markers in routine clinical use in contrast to other malignancies such as ovarian (cancer antigen 125), prostate (prostate-specific antigen), or pancreatic cancers (carbohydrate antigen 19-9 (CA19-9)) [10]. The high mortality and morbidity rates of brain tumors indicate the need for novel and high-throughput tools to identify these disorders. Protein glycosylation is a covalent linkage of a carbohydrate chain to a polypeptide backbone through an asparagine (N-glycosylation) or serine/threonine (O-glycosylation), reportedly altered in several inflammatory [11] and cancerous diseases [12]. Generally, the main glycosylation alterations in cancer are the increased fucosylation and higher degree of branching due to altered fucose metabolism and N-acetyl-glucosamine-transferase activity [13]. Fucose residues can normally occur on a wide range of molecules as most IgG glycans are α1-6 fucosylated, while ABO blood group antigens are fucosylated glycans as well [14]. Alterations in GDP-fucose synthesis, fucosyl-transferase, and fucosidase activity can lead to aberrant fucosylation, which has been identified as a major signature of uncontrolled cell proliferation, tumor cell invasion, and metastasis [14]. Increased branching of glycans—a shift from bi-antennary to tri- and tetra-antennary structures—is also found to be correlated with cellular transformation and progression [15]. The detection of increased fucosylation and branching by analytical techniques have provided novel biomarkers in cancerous diseases such as α-fetoprotein fucosylation in hepatocellular carcinoma and highly fucosylated haptoglobin in pancreatic cancer, suggesting a potential supportive tool for the detection of malignancy [16].

In this study, serum N-glycosylation of patients diagnosed with brain tumors (meningioma Grade I., glioblastoma Grade IV., metastasis) was analyzed by hydrophilic interaction liquid chromatography (HILIC) and compared to healthy controls. The N-glycosylation patterns were quantified by fluorescence and identified by mass spectrometric detection in combination with exoglycosidase digestions. Multivariate data analysis provided a clear separation of different disease groups, along with the main contributor glycan species.

## 2. Materials and Methods 

### 2.1. Chemicals

Formic acid, ammonium-hydroxide, acetic acid, acetonitrile, picoline borane, procainamide-hydrochloride, and dimethyl sulfoxide were purchased from Sigma-Aldrich (St. Louis, MO, USA). PNGase F was provided by Asparia Glycomics (San Sebastian, Spain). An exoglycosidase sequencing kit was obtained from New England Biolabs (Ipswich, MA, USA).

### 2.2. Patient Samples

Serum samples from 33 sex- and age-matched healthy controls (average age 66.2) and 33 patients diagnosed with brain tumors (11 meningioma Grade I, 11 glioblastoma Grade IV, 11 metastases, average age 67.5) were collected in the Department of Neurosurgery at the Borsod-Abaúj-Zemplén County Center Hospital and University Teaching Hospital (Miskolc, Hungary). Informed consent was obtained from all subjects in agreement with the Declaration of Helsinki. All methods were carried out in accordance with relevant guidelines and regulations. All experimental protocols were approved by the Regional Research Ethics Committee of the Borsod-Abaúj-Zemplén County Center Hospital and University Teaching Hospital (ethical approval number: RKEB/IKEB-G-102-102-2018).

### 2.3. N-Glycan Release, Labeling, and Clean-Up

Glycan release was performed using 5 µL of human serum, according to the PNGase F denaturing deglycosylation protocol of Asparia Glycomics (San Sebastian, Spain). The released glycans were labeled by the addition of 10 μL 0.3 M procainamide and 300 mM picoline borane in 70%/30% of dimethyl sulfoxide/acetic acid incubating for 2 h at 65 °C. The purification of labeled glycans was performed by hydrophilic magnetic iron nanoparticles, as described in our recent publication [17].

### 2.4. UPLC-FLR-MS Analysis

The fluorescently labeled and purified N-glycans were analyzed using a Waters Acquity ultra-performance liquid chromatography instrument with fluorescence and single quad mass spectrometric detection. Empower 3 chromatography software (Waters, Milford, MA, USA) was used to control the system. Chromatographic separations were performed by a Waters BEH Glycan column, 100 × 2.1 mm i.d., 1.7 μm particles, with a gradient of 72–55% acetonitrile in 42 min at a 0.4 mL/min flowrate using 50 mM ammonium formate pH 4.4 as the mobile phase. Samples were made up of 75%/25% acetonitrile/ water, and 5 μL was injected using partial loop injection in all runs. The sample manager temperature was 15 °C, and the column temperature was 60 °C during each analysis. The fluorescence detection excitation and emission wavelengths were λex = 309 nm and λem = 359 nm. In the MS setup, the electrospray voltage was 2.2 kV, with a desolvation temperature of 120 °C using a 500 L/hr desolvation gas flow. Mass spectra were acquired using a positive ionization mode over the range of 600–2000 *m*/*z*. Centroid data was acquired with a scan rate of 500 Da/s, and the MS target resolution was 0.5 Da.

### 2.5. Data Analysis

Each patient sample was analyzed in triplicate, and all chromatograms were integrated by Empower 3 chromatography software (Waters, Milford, MA, USA). The generated data were analyzed in Past 4.02 [18], performing linear discriminant analysis of the different patient groups. Kruskal–Wallis tests of individual glycan structures were obtained by IBM SPSS Statistics 25. The mass calculation of the individual glycan structures was performed in GlycoWorkBench. Glycan nomenclature was used, as described by Harvey et al. [19].

## 3. Results and Discussion

### 3.1. Magnetic-Nanoparticle-Based Sample Preparation

Glycans lack fluorescent moieties; therefore, several fluorophore tags have been used for glycan analysis in recent years, such as 2-aminobenzamide (2-AB), 2-anthranilic acid (2-AA), and 1-aminopyrene-3,6,8-trisulfonic acid (APTS) [20]. Latest advances in glycoanalytics have revealed significantly different fluorescence signal intensities and MS ionization efficiencies between the different labels [21]. To measure these disparities, equal amounts (10 μg) of dextran ladders were reductively aminated by 2-AB, 2-AA, and procainamide followed by HILIC-UPLC analysis with fluorescence and MS detection. As visualized in Appendix A, procainamide labeling provides more than 12 times higher fluorescence signal intensities and 2 magnitudes higher MS ionization efficiency in positive mode (Appendix A). Mass spectrometric measurements were also performed in negative mode, where the procainamide-labeled sample also provided the highest intensity (data not shown). These results are in agreement with Keser et al., suggesting higher detection sensitivity by procainamide labeling compared to 2-AB [21]. To further develop this improvement, we examined the compatibility of our recently developed magnetic nanoparticle-based sample clean-up with procainamide-labeled glycans. Firstly, glycans were released from 5 µL of human serum and reductively aminated by 2-AB, 2-AA, and procainamide, followed by a magnetic-nanoparticle-based sample purification. All of the nanoparticle synthesis and glycan purification steps were identical to the protocol published recently [17]. As visualized in Figure 1, the inhouse-synthetized magnetic nanoparticles could be used for the purification of glycans labeled with negatively charged (anthranilic acid), neutral (aminobenzamide), and positively charged (procainamide) fluorescent tags as well. Based on this result, procainamide labeling, with increased detection sensitivity and inhouse-synthetized magnetic nanoparticle-based sample clean-up, was used to analyze patient samples by quantitative HILIC-UPLC/FLR. 

### 3.2. Structural Annotation of Serum N-Glycome

To identify potential disease-associated alterations, serum N-glycans have been annotated based on their *m*/*z* values and subsequent exoglycosidase digestions (Figure 2). The annotated structures were mainly bi-, tri-, and tetra-antennary glycans with various degrees of sialylation. As shown in Figure 2, each individual fluorescence peak (Figure 2A) had a main correspondent *m*/*z* value in the total ion chromatogram (TIC; Figure 2B) of the MS due to the high-resolution separation of the UPLC. The main ions of neutral and monosialylated structures were mostly doubly-charged (M+2H)^2+^, although lower amounts of singly-charged ions (M+H)^+^ were also detected. Similarly, the structures with 2, 3, and 4 sialic acids have presented mainly doubly- (M+2H)^2+^ but also triply- (M+3H)^3+^ charged ions, improving the structural annotation of the individual peaks. Structural isomers have also been found as multiple peaks and have shown the same *m*/*z* values (1082.78^3+^, 1082.41^3+^); thus, exoglycosidase digestions have been performed (Appendix A). After removing the sialic acids by sialidase digestion (Appendix A), we also found multiple peaks with the same masses and charge states (1186.82^2+^, 1186.75^2+^), which were identified as possible isomers due to the different linkages of fucose residues. This has been confirmed by separate α1-6 and α1-3/4 fucosidase digestions, as shown in Appendix A. Based on these results, the first ion of 1186.82^2+^, at the retention time of 24.5 min, was identified as an α1-6-fucosylated tri-antennary structure, while the one at 25.5 min was an α1-3/4 fucosylated glycan structure. By the addition of 3 sialic acids to these structures, we identified the corresponding glycan structures of 1082.78^3+^ and 1082.41^3+^ as FA3G3S3 and A3FG3S3. These results are in agreement with previous studies, where serum glycans have been annotated with HILIC-UPLC and exoglycosidase sequencing [22,23,24].

### 3.3. Data Analysis

During the data analysis, 46 glycan structures were quantified from 66 participants (33 patients, 33 healthy controls); 30 showed significantly altered relative areas, as listed in Appendix A. One of the main trends comparing the controls to all brain cancer patients was the lower level of neutral and biantennary structures and the subsequently higher ratio of tri- and tetra-antennary structures with terminal sialic acids in cancer patients (Appendix A). To determine the order of importance is challenging when the number of significant alterations is high; thus, a linear discriminant analysis was performed on the relative peak areas. As shown in Figure 3, the different disease groups can be well-separated based on their glycan distributions, with a slight overlap of the groups of glioblastoma and metastatic patients. The contribution of the individual structures is also visualized on this biplot, suggesting the main alterations by which the groups can be isolated. It can be seen that the separation of the control group is due to the altered ratio of biantennary neutral structures (as described above), while the patients with meningioma had altered sialylation and the glioblastoma-metastatic patients showed altered fucosylation.

To verify the credibility of the biplot generated by linear discriminant analysis, pairwise comparisons were performed to identify some of the main significant alterations between the different groups (Figure 4). Representing the main alterations, FA2G2 (Figure 4A), A2G2S1 (Figure 4B), and A3FG3S3 (Figure 4C) were selected to show the typical distribution of biantennary neutral, sialylated but nonfucosylated, and sialylated–fucosylated structures. As shown in Figure 4A, FA2G2 was significantly higher in the control group compared to all the different groups of brain cancer patients. This distribution was typical for all biantennary neutral glycans, mainly contributing to the isolation of the control group. Increased sialylation was found to be responsible for the separation of the patients with meningioma, which was found to be significant compared to the controls and to the patients with metastasis as well in the case of A2G2S1 (Figure 4B); however, A2G2S2 and A3G3S3 also showed the very same deviation (data not shown). The patients with glioblastoma and metastasis have shown increased fucosylation on bi-, tri-, and tetra-antennary sialylated structures. As shown in Figure 4C, α1-3/4-linked A3FG3S3 was significantly increased in patients with metastasis compared to the controls and patients with meningioma. 

To our knowledge, this is the first study analyzing serum N-glycome of patients with different types of brain tumors. The main alterations identified in this study were the increased sialylation and higher branching in cancer patients compared to healthy controls.

Sialylation—the presence of negatively charged terminal N-acetyl-neuraminic acid—is involved in several biological processes such as signal transduction, cell adhesion, and immune modulation [25]. The overexpression of sialic acids on the surface of cancer cells has, reportedly, a crucial impact on their metastatic potential due to created repulsion by the negative charges [26]. Hu et al. reported increased sialylation and higher branching in NLF neuroblastoma cells by total glycomic analysis, revealing that the altered glycomic profile can be associated with the progression of the cells and their aggressivity [27]. Elevated total serum sialic acid level was found to be correlated with the presence of CNS tumors in men, as reported by Gatchev et al. [28], while Gökmen et al. have also reported higher serum sialic acid levels in patients with lung cancer compared to healthy individuals [29]. These findings are all in agreement with our results, as there was clear evidence of sialylation increase in cancer patients.

Fucosylation contributes to various biochemical events involving pathogen–host interactions, leukocyte extravasation, and signaling. Aberrant fucosylation is a major signature of malignant transformation as fucosylation has been found to be involved in cell proliferation, invasion, and metastasis [14]. Kumar et al. reported that serum fucose levels are significantly increased in patients with oral cancer, correlating with the histopathological grading [30]. A significant relationship between fucose accumulation and advanced stage in colorectal cancer has been reported by Osuga et al. [31], while Zipin et al. have identified α1-3/4 fucosyl-transferase as a potential regulator of metastasis in colorectal adenocarcinoma [32]. By analyzing serum glycosylation, altered fucosylation has been reported in prostate, gastric, breast, and oesophageal cancers, revealing higher proportions of sialyl Lewis x epitope (sLe^x^, a tetrasaccharide composed of an α1-3-linked fucose to an N-acetyl-glucosamine and connected to a galactose, terminating in sialic acid) [16]. This alteration is in agreement with our findings, as the elevated A3FG3S3 (the fucosylated arm is a sLe^x^ epitope) was the main contributor to the separation of the different disease groups. Our results demonstrate that decreased levels of neutral (biantennary) glycans and subsequent increase in sialylation on highly branched structures (tri- and tetra-antennary) might be early signs of uncontrolled cell proliferation, while the further extension of these highly branched and sialylated structures, with α1-3/4-linked fucose (creating sLe^x^), can justify malignant transformation and metastasis. These findings are also supported by a recent study claiming stage-associated serum glycosylation alterations in patients with nonsmall lung cancer, where increased sialylation was typical in early stages and higher fucosylation was observed in advanced stages [33].

It has to be clarified that the alterations presented in this study are not unique to brain tumors, as increased fucosylation has been reported in several types of cancers; moreover, further evaluation is essential to demonstrate the diagnostic value on a larger number of sample sets.

## Figures and Tables

**Figure 1 life-11-00029-f001:**
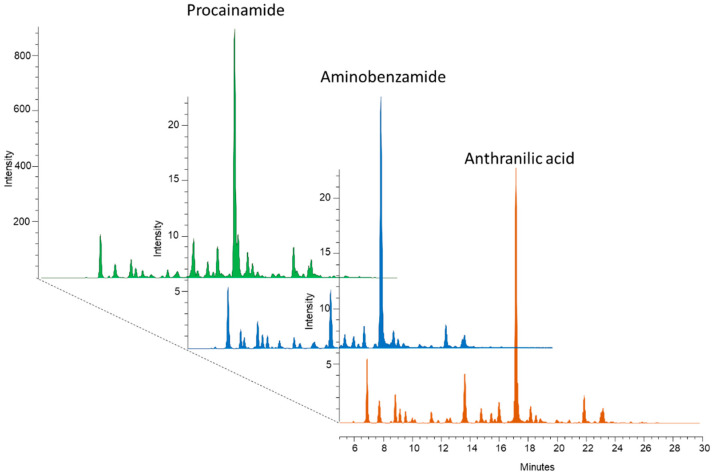
Magnetic nanoparticle purified serum N-glycans labeled by procainamide, 2-AB, and 2-AA.

**Figure 2 life-11-00029-f002:**
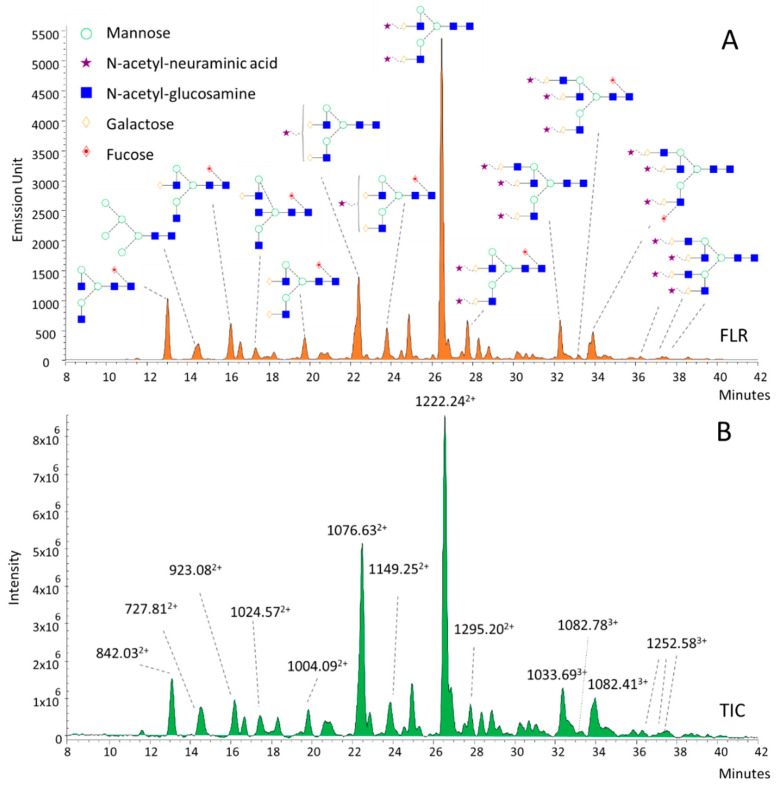
HILIC-UPLC profile of procainamide-labeled serum N-glycans by fluorescence (**A**) and mass spectrometric detection (**B**).

**Figure 3 life-11-00029-f003:**
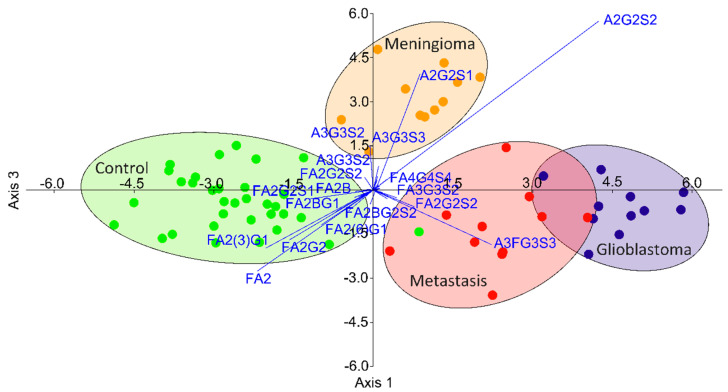
Separation of patients with different brain cancers by linear discriminant analysis based on their serum N-glycan distribution.

**Figure 4 life-11-00029-f004:**
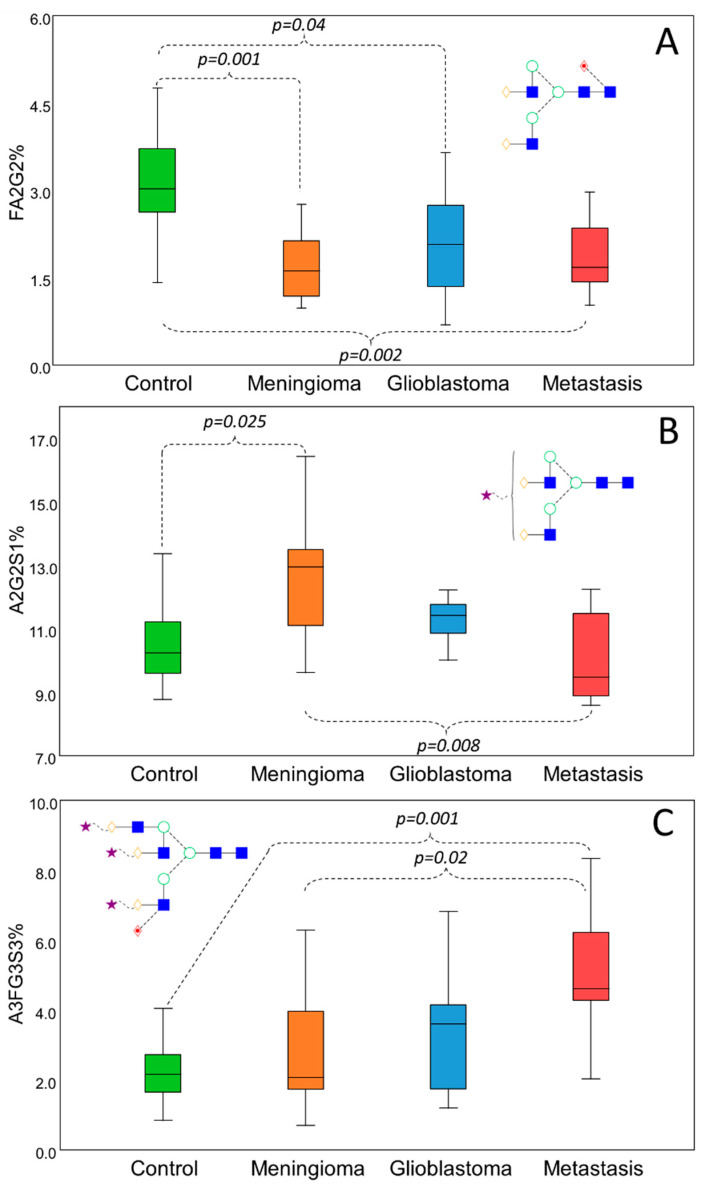
Significant serum N-glycosylation alterations in different brain cancers by Kruskall–Wallis test. Decreased level of biantennary neutral glycans in cancer patients (**A**) and subsequent increase in sialylation (**B**) and fucosylation (**C**) compared to the controls.

## Data Availability

The data presented in this study are available on request from the corresponding author.

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
