# Peer review of "The Analysis of Human Serum N-Glycosylation in Patients with Primary and Metastatic Brain Tumors"

_life, 2021, doi:10.3390/life11010029_

Round 1

Reviewer 1 Report

This study reports a strategy to purify N-glycans labeled procainamide via iron nanoparticles. Then, the authors purify and analyze serum N-glycome of patients with different types of brain tumors via UPLC-FLR-MS. According to their findings, they suggested that serum glycosylation monitoring could improve the detection of malignancy. However, there are some major issues need to be addressed before this draft can be considered for publication in “Life”. Therefore, a “major revision” is required.

  1. The characterization of nanoparticle is suggested to be provided;
  2. To show the advantage of N-glycans purification via iron nanoparticle (e.g., for comparison of the recovery via nanoparticles), the UPLC-FLR-MS data of serum N-glycans labeled with individual dyes is required;
  3. As the iron nanoparticles carry negative charge, the free dye will affect the reading. Therefore, the data of nanoparticle interacts with free dye is required as well;
  4. There are too much discussion has been included in “Results” section. Either combine “results” and “discussions” or move these statements into “discussion”.
  5. Please verify which Statistical analysis was applied in Figure 4? Does p=0.002 stand for “**”. However, it seems that there are not any difference between groups. Please explain.
  6. Some errors, such as lane 35 “tumors” should be “tumor”, lane 43 “cancers” should be “cancer”, lane 121 “ionisation” should be “ionization”, etc. should be corrected. 

Author Response

Reviewer 1

This study reports a strategy to purify N-glycans labeled procainamide via iron nanoparticles. Then, the authors purify and analyze serum N-glycome of patients with different types of brain tumors via UPLC-FLR-MS. According to their findings, they suggested that serum glycosylation monitoring could improve the detection of malignancy. However, there are some major issues need to be addressed before this draft can be considered for publication in “Life”. Therefore, a “major revision” is required.

  1. The characterization of nanoparticle is suggested to be provided;

Reply1: The nanoparticles have been characterized in our previous study(https://doi.org/10.3390/nano9101480).

  1. To show the advantage of N-glycans purification via iron nanoparticle (e.g., for comparison of the recovery via nanoparticles), the UPLC-FLR-MS data of serum N-glycans labeled with individual dyes is required;

Reply 2: The sample recovery has been also analysed in the previous study as you can see in the picture below. This figure is the supplementary figure 2 of https://doi.org/10.3390/nano9101480. Please note that the UPLC-FLR-MS intensities are depending on the nature of the fluorescent dye so comparing different dyes purified by any methods does not reflect the efficiency of the purification. It might be affected but it is mainly up to the ionization efficieny and fluorescent properties of the dye. Our aim was to show our nanoparticle based approach is suitable for glycan purification labeled with different fluorescent tags and as it shown in the supplementary figure 1-2 procainamide provides the highest fluorescence and mass spectrometric signals.

  1. As the iron nanoparticles carry negative charge, the free dye will affect the reading. Therefore, the data of nanoparticle interacts with free dye is required as well;

Reply 3: Iron oxides are likely to form hydrogen donor interactions with hydroxyl groups and adsorb polar compounds. As glycans have numerous hydroxyl groups they can form hydrogen bonds with iron oxides. This is supported by the fact that the acetonitrile concentration for binding and washing steps had to be increased (water content decreased) thus iron oxides formed hydrogen donor interaction with the OH group of glycans instead of water. The addition of water as an eluent resulted in the dissociation of hydrogen bonds, thus the adsorbed sugars could be isolated from supernatant.

  1. There are too much discussion has been included in “Results” section. Either combine “results” and “discussions” or move these statements into “discussion”.

Reply 4: The results and discussion have been combined.

  1. Please verify which Statistical analysis was applied in Figure 4? Does p=0.002 stand for “**”. However, it seems that there are not any difference between groups. Please explain.

Reply 5: The applied stratistical analysis was Kruskall-Wallis test. As you can see below it makes pairwise comparisons between the groups and shows if any of the groups is significantly different from another one. This figure corresponds to the manuscript Figure 4C.

  1. Some errors, such as lane 35 “tumors” should be “tumor”, lane 43 “cancers” should be “cancer”, lane 121 “ionisation” should be “ionization”, etc. should be corrected.

Reply 6: Thank you for your suggestions. The errors have been corrected.

Reviewer 2 Report

In this manuscript, the authors conducted a LC-FL-MS glycomic analysis of sera from brain cancer patients. A total of 33 patient sera and 33 controls were analyzed. The significant changes of sialylation and fucosylation in brain patient sera were observed, as well as the up- or down-regulations of several individual glycans. The outcome of this paper provided more information about the role glycans may play in brain cancer progression. However, my major concern is the glycan identification. Both FL and low-resolution MS could not be used for a promising structural elucidation. Without MS2, it is hard to believe the glycan structures presented in the manuscript, especially for some positional isomers and bisecting structures. Please add identification figures including MS spectrum for these glycans.

  1. In Figure 1, please add y-axis
  2. In the method part, please add the MS resolution settings.
  3. In Supplementary Figure 5, it is hard to tell the difference between green and orange bars because the y-axis is from 0 to 100%. Please change it to another format such as butterfly chart or conventional bar graph with two bars next to each other.
  4. Supplementary Table 1, the authors used “,” in the middle of numbers. Could it be “.”? Please double check. Please add theoretical m/z, detected m/z, and accuracy (ppm) to the table.
  5. Which detection method was used for the quantitation? MS or FL? If it was FL, the quantitation may not be accurate because of the overlapping glycan peaks. If it was MS, please present the advantages of using FL in this study.
  6. Since this study is related to clinical samples, please add ROC and AUC to emphasize the significance of the results.

Author Response

Reviewer 2

In this manuscript, the authors conducted a LC-FL-MS glycomic analysis of sera from brain cancer patients. A total of 33 patient sera and 33 controls were analyzed. The significant changes of sialylation and fucosylation in brain patient sera were observed, as well as the up- or down-regulations of several individual glycans. The outcome of this paper provided more information about the role glycans may play in brain cancer progression. However, my major concern is the glycan identification. Both FL and low-resolution MS could not be used for a promising structural elucidation. Without MS2, it is hard to believe the glycan structures presented in the manuscript, especially for some positional isomers and bisecting structures. Please add identification figures including MS spectrum for these glycans.

Author response: We understand the MS2 would be the most comprehensive structural elucidation although as our low resolutiom mass spectrometer cannot provide MS2 we have applied multiple exoglycosidase digestions to identify the glycan structures which is widely accepted since years in analytical glycomics. Please note that we are also referencing previous papers dealing with procainamide labeled glycans and our structure list is in agreement with those. More importantly we are not claiming that any of our identified structures could be potential biomarkers of the disease, we are only claiming that the altered serum N-glycan distribution can help to improve disease identification.

  1. In Figure 1, please add y-axis

Reply 1: It has been added.

  1. In the method part, please add the MS resolution settings.

Reply 2: It has been added.

  1. In Supplementary Figure 5, it is hard to tell the difference between green and orange bars because the y-axis is from 0 to 100%. Please change it to another format such as butterfly chart or conventional bar graph with two bars next to each other.

Reply 3: The chart has been changed. Thank you for the suggestion.

  1. Supplementary Table 1, the authors used “,” in the middle of numbers. Could it be “.”? Please double check. Please add theoretical m/z, detected m/z, and accuracy (ppm) to the table.

Reply 4: Thank you for your suggestion. All the required changes have been made.

  1. Which detection method was used for the quantitation? MS or FL? If it was FL, the quantitation may not be accurate because of the overlapping glycan peaks. If it was MS, please present the advantages of using FL in this study.

Reply 5: We have used fluorescence detection for the quantitation as hidrophilic interaction liquid chromatography is a widely accepted method since years providing high resolution separation of glycans. Please note that if glycan peaks are overlapping in the fluorescence they will overlap in the MS as well. Obviously MS based quantitation can be an option but for that high resolution mass spectrometers are preferred. Our aim was to show that we can detect malignany from 5µL of serum which needs highly sensitive detection methods such as fluorescence detection.

  1. Since this study is related to clinical samples, please add ROC and AUC to emphasize the significance of the results.

Reply 6: ROC and AUC are used in large-scale biomarker studies. Please note that we only had 11 patients/disease groups thus we would like to avoid from such statistics. We do not claim that the reported alterations are biomarkers. We only show potential of glycan analysis in the detection of malignancy which could improve current diagnostic procedures.

Round 2

Reviewer 1 Report

Most comments have been addressed. Now it meets the requirement for publication.

Author Response

Reviewer 1:

Most comments have been addressed. Now it meets the requirement for publication.

Reply: Thank you for your comment.

Reviewer 2 Report

In the revised manuscript, the authors well addressed the reviewer's comments. The manuscript now meets the requirement of publication. My additional suggestion is to add error bars on Supplementary Figure 5.

Author Response

Reviewer 2:

In the revised manuscript, the authors well addressed the reviewer's comments. The manuscript now meets the requirement of publication. My additional suggestion is to add error bars on Supplementary Figure 5..

Reply: Thank you for your suggestion. The Supplementary Figure 5 has been modified.
